# Anti-Cytokine Therapy to Attenuate Ischemic-Reperfusion Associated Brain Injury in the Perinatal Period

**DOI:** 10.3390/brainsci8060101

**Published:** 2018-06-07

**Authors:** Clémence Disdier, Xiaodi Chen, Jeong-Eun Kim, Steven W. Threlkeld, Barbara S. Stonestreet

**Affiliations:** 1Department of Pediatrics, Women & Infants Hospital of Rhode Island, The Alpert Medical School of Brown University, Providence, RI 02905, USA; CDisdier@wihri.org (C.D.); XChen@wihri.org (X.C.); jkim62@northwell.edu (J.-E.K.); 2Department of Neuroscience, Regis College, Weston, MA 02493, USA; Steven.threlkeld@regiscollege.edu

**Keywords:** brain, cytokines, ischemia reperfusion, neuro-inflammation, ovine fetus, monoclonal antibodies

## Abstract

Perinatal brain injury is a major cause of morbidity and long-standing disability in newborns. Hypothermia is the only therapy approved to attenuate brain injury in the newborn. However, this treatment is unfortunately only partially neuroprotective and can only be used to treat hypoxic-ischemic encephalopathy in full term infants. Therefore, there is an urgent need for adjunctive therapeutic strategies. Post-ischemic neuro-inflammation is a crucial contributor to the evolution of brain injury in neonates and constitutes a promising therapeutic target. Recently, we demonstrated encouraging neuroprotective capacities of anti-cytokine monoclonal antibodies (mAbs) in an ischemic-reperfusion (I/R) model of brain injury in the ovine fetus. The purpose of this review is to summarize the current knowledge regarding the inflammatory response in the perinatal sheep brain after I/R injury and to review our recent findings regarding the beneficial effects of treatment with anti-cytokine mAbs.

## 1. Introduction

Hypoxic-ischemic (HI) brain injury is the most significant common neurologic problem occurring during the perinatal period [1]. HI injury can result in significant mortality and long term neurological sequelae including cognitive, behavioral, and intellectual deficits [2,3,4,5]. Therapeutic hypothermia or targeted temperature management is the only effective neuroprotective strategy available to attenuate secondary cerebral insults associated with hypoxic-ischemic encephalopathy (HIE). However, hypothermia is only approved to treat newborns, who are 36 weeks of gestation or greater with HIE. In addition, hypothermia is only partially neuroprotective and surviving neonates with moderate to severe HIE remain at risk of dying or developing significant neurodevelopmental impairment, emphasizing the need for additional adjunctive therapeutic strategies [6,7,8,9]. 

Multiple complex processes occur in the brain after ischemic-reperfusion (I/R) related injury including excitotoxicity, oxidative stress, free radical and inflammatory mediator release, and blood-brain barrier (BBB) dysfunction [10,11,12]. The responses to I/R evolve over time, are interdependent, and result in neuronal and glial injury and cell death. Post-ischemic inflammation represents a critical component in the evolution of brain injury [13,14,15,16]. The inflammatory response begins within hours after an ischemic insult but can last from days up to weeks after the insult. Pro-inflammatory cytokines including tumor necrosis factor (TNF-α), interleukin-1β (IL-1β), and interleukin-6 (IL-6) can be released within the systemic circulation and locally within the central nervous system (CNS) parenchyma. Elevated cytokines within the brain parenchyma can originate locally from stimulated intrinsic cells within the brain including activated cerebrovascular endothelium, from infiltrated cells having originated in the systemic circulation and/or by crossing the normal or injured BBB. Cytokines can then stimulate or amplify inflammatory cascades within the CNS. Activation of cytokine signaling pathways can result in sustained inflammation that could accentuate the ischemic damage. Inflammation has been shown to affect brain development with long-lasting consequences predisposing to neurologic disorders [13,17,18]. Strategies to attenuate secondary damage resulting from CNS inflammation could have a potentially wider therapeutic window compared with therapies that target primary damage immediately after an ischemic insult. Consequently, accumulating evidence suggests that targeting pro-inflammatory cytokines could represent a potentially important neuroprotective strategy to treat perinatal HI injury [10,19,20,21,22]. However, caution needs to be exercised when considering cytokines as therapeutic targets because their roles remain controversial as they have both favorable and detrimental effects within the CNS [23,24]. The immune system is a dynamic contributor to wound healing and at least some inflammation is critical in the early stages of CNS injury to remove damaged tissue and promote tissue remodeling [25]. Moreover, maternal cytokines that cross the placenta are important to the establishment and maintenance of pregnancy, as well as for normal fetal development [26,27,28,29,30,31].

The sheep fetus has been widely used to examine many aspects of brain development [32,33,34,35] as the neurodevelopment of the immature sheep brain has many similarities to those of premature neonates [36,37,38,39]. Sheep pregnancy at full term is considered to be 145–150 days gestation. The fetal sheep brain between 94 and 96 days of gestation is considered similar to that of the preterm human infant between 24 and 28 weeks of gestation, whereas the fetal sheep brain at approximately 135 days of gestation is similar to that of a full term infant [32,39]. Therefore, the ovine fetus represents a very useful well established translational model to study HI injury in the preterm and full-term brain. We have formerly described white matter and cortical injury resulting from bilateral carotid artery occlusion in the ovine fetus [40]. In this context, we have examined the effects of *in utero* brain ischemia on inflammatory responses in order to develop therapeutic strategies using neutralizing anti-cytokine monoclonal antibodies (mAbs). We produced and purified specific and sensitive mouse anti-ovine IL-1β and -IL-6 mAbs for use in fetal sheep that effectively neutralized the effects of the corresponding proteins [41,42,43]. The purpose of this review is to summarize our findings in the ovine fetal model of I/R induced brain injury and our advances in the progress with two potentially neuroprotective anti-cytokine antibodies.

## 2. Inflammation Associated with Ischemic-Reperfusion Injury in the Ovine Fetal Brain

Inflammation with the release of pro-inflammatory cytokines is central to the progression of the brain injury after HI [13,14,15,16]. Both the peripheral and CNS immune systems contribute to the activation of brain inflammation after HI. IL-1β and IL-6 have been identified as important pro-inflammatory cytokine intermediaries involved in inflammatory responses after HI [44]. Evidence suggests that inflammatory proteins such as IL-1β can alter neuronal function and synaptic transmission in acute as well chronic inflammatory conditions [45,46]. 

### 2.1. Cytokine Expression in the Developing Brain

IL-1β and IL-6 exhibit differential patterns of regional expression in the ovine fetal brain during development [47]. The constitutive expression of IL-1β increased in the cerebral cortex from early in fetal life (87–90 days of gestation) up to near term gestation (135–137 days of gestation). The expression of IL-6 also increased in white matter during fetal development and exhibited upregulation in the cerebral cortex at 122–127 days of gestation. In addition, gestational age- and brain region-dependent variations in the expression patterns of IL-1β and IL-6 were observed with increasing gestation in the brains of the pregnant ewes. The increases in IL-1β and IL-6 expression in the CNS of the ewes during gestation suggest that cytokines could contribute to interactions between the maternal immune and reproductive systems, and that these interactions could include functions within the normal maternal brain during pregnancy [47]. In summary, these findings suggest a key role for cytokines during normal fetal brain development, during normal pregnancy in the maternal brain, and in pregnancy maintenance [26,27,28,29,30,31].

Changes in IL-1β, IL-6, and the high-mobility group box-1 (HMGB1) cytokines were also observed after ischemia in the brains of the fetal sheep [47,48,49,50,51,52]. Ischemia resulted in increases in the expression of IL-1β and IL-6 in the white matter and of IL-1β in the cerebral cortex after 30 min of bilateral carotid artery occlusion followed by 48 h or 72 h of reperfusion in utero [47]. HMGB1 is a nuclear protein, which is translocated from the nucleus to the cytoplasm and released after ischemia in the brain of adult rodents, thereby augmenting the inflammatory response [53,54]. We have recently reported that ischemia results in the neuronal translocation of HMGB1 from the nuclear to cytoplasmic compartment in the brain of fetal sheep [48]. The translocation may facilitate the action of HMGB1 as a pro-inflammatory cytokine and serve to accentuate HI injury in the developing brain. Therefore, several pro-inflammatory cytokines exhibit upregulation after ischemia in the brain of fetal sheep. These observations emphasize that the cytokines could be important mediators of HI-related brain injury during the perinatal period. We cannot be certain of the specific cellular origin of the cytokines in our studies. However, microglia and astrocytes are the two major reactive glial cell types that play significant roles during CNS injury [13]. Pro-inflammatory cytokines are produced by both intrinsic and infiltrating immune cells. Circulating immune cells including neutrophils infiltrate the brain parenchyma to further exacerbate the inflammatory response and consequently exacerbate brain injury. The infiltration of leukocytes is facilitated by chemokine secretion and the expression of adhesion molecules at the endothelial cell surface. Moreover, neutrophils that aggregate in the cerebral microvasculature can interfere with cerebral circulation and potentially further exacerbate brain injury [16]. Although the upregulation of pro-inflammatory cytokines in the brain could result from increases in local production by activated glia cells, neurons, and microvascular endothelial cells, it could also result from increases in systemic circulating levels of cytokines along with increased transport across the BBB after ischemia. In the ovine model of repetitive umbilical cord occlusions, similar inflammatory responses were observed in the fetal brain. IL-1β expression was increased in blood up to 24 h after exposure to repetitive umbilical cord occlusions and microglia counts were increased in the white matter and hippocampus at 24 h [55]. Other studies also demonstrated the central activation of microglia associated with systemic inflammation and the influx of neutrophils into the brain after 25 min of umbilical cord occlusion in fetal sheep [56,57,58,59].

### 2.2. Blood-Brain Barrier Dysfunction

The BBB is primarily composed of microvascular endothelial cells that comprise the brain capillary walls but it has very specific features that result from close interactions with other brain cells that constitute the neurovascular unit including pericytes, astrocytes end foot processes, and basement membranes [60,61]. The BBB regulates the exchanges between the brain and blood to provide an optimal environment for neuronal activity. Restriction of the entry of low molecular weight molecules by paracellular diffusion is mainly due to tight junction (TJ) complexes that seal neighboring microvascular endothelial cells [62]. Passage of essential nutrients into the brain is facilitated by a wide variety of transporters and receptors that allow for the uptake of some large molecules and small nutrients into the brain. These include the glucose transporter GLUT1 and the Solute Carriers (SLC) for many amino acids [60]. Several studies, including our work on fetal sheep, have described BBB dysfunction resulting in vasogenic edema after HI [49,50,51,63,64,65]. We have examined the BBB permeability by calculation of the blood-to-brain transfer constant measured with α-aminoisobutyric acid in the fetal sheep brain [66]. We demonstrated that BBB permeability was increased from 4 h up to 48 h after brain ischemia, indicating the leakage of potential blood-borne neurotoxic constituents into the brain parenchyma [51]. The expression of four TJ proteins (occludin, claudin 5, and accessory proteins zonula occludens ZO-1 and -2) was decreased 4 h after ischemia, consistent with the increases in BBB permeability. However, we observed a partial recovery of BBB function, as well as a return toward control levels for TJ protein expression, beginning at 24 h after reperfusion. This suggests that BBB leakage is a transient phenomenon in the first few hours after the insult, followed by partial restoration of the barrier function. Morphological analysis of the fetal cerebral cortex also revealed that neovascularization occurs within 72 h after ischemia in the fetal brain [67]. Neovascularization was associated with increases in cerebral cortical basic fibroblast growth factor and astrocytic proliferation. Astrocytes represent vital components of the neuro-vascular unit [68,69]. This further supports the contention that the cerebral vasculature begins to recover within days after an ischemic insult in the fetal brain. 

Perturbations in BBB function can also be mediated by changes in pro-inflammatory cytokines [70,71]. Cytokines bind to their receptors at the endothelial cell surface and activate signaling pathways to enhance endothelial abnormalities, leading to modifications in the physical barrier. In this regard, we demonstrated that exposure to IL-6 protein for 24 h in vitro reduced TJ protein expression (claudin-5 and occludin) in cerebral cortical microvessels from yearling and adult sheep [72]. However, the specific signaling pathways involved in such regulation remain to be determined.

Likewise, microvascular endothelial cells are active participants in the neuro-immune response by the transport of circulating pro-inflammatory cytokines [73,74,75,76]. There is a considerable quantity of clinical evidence supporting the concept that elevated circulating pro-inflammatory cytokines predispose to the development of brain damage in neonates [77,78]. In order for inflammatory substances produced in the systemic circulation to reach the fetal brain, they need to cross the BBB of the fetus [77]. Disorders such as systemic inflammation, sepsis, necrotizing enterocolitis, and mechanical ventilation in the premature infant, which increase systemic pro-inflammatory cytokines, also result in future brain injury in premature infants [79,80,81]. Although it has long been postulated that systemic cytokines could gain access to the fetal brain by crossing the BBB, there was very little experimental evidence to support this contention until our recent report [52]. We have shown that systemic IL-1β crosses the intact BBB and that ischemia with reperfusion for four hours facilitates the entrance of systemic IL-1β into brain parenchyma [52]. These findings confirmed our earlier data showing that ovine IL-1β, as well as IL-6, was able to cross the murine BBB by the use of a saturable transport mechanism [73]. The increase in cytokine transport across the fetal BBB after ischemia might result from increases in paracellular BBB leakage after ischemia and/or increases in receptor mediated transcellular transport [52]. Consequently, the changes in the BBB physiology after I/R increase the availability of pro-inflammatory cytokines within the brain parenchyma [52]. Taken together, our observations in the ovine brain emphasize that BBB dysfunction represents a central constituent of I/R related injury in the fetus, pro-inflammatory cytokines predispose to the endothelial barrier dysfunction, and pro-inflammatory cytokines can cross the intact and injured BBB in the fetus. Therefore, we have postulated, based upon the findings described above, that neutralizing the detrimental effects of pro-inflammatory cytokines at the BBB and in the cerebral parenchyma could represent a potentially beneficial strategy to attenuate perinatal brain injury after HI. The multiple interactions between inflammatory mediators and BBB abnormalities after HI insult are schematically represented in Figure 1.

## 3. Anti-Cytokines Antibodies as a Potential Neuroprotective Strategy

### 3.1. Anti-Cytokines Strategy

The significant contribution of pro-inflammatory cytokines, especially IL-1β, in HI-induced brain injury has been increasingly recognized and represents an attractive therapeutic target. The infarct volume in an adult rat model of ischemic brain injury was attenuated after administration of a recombinant IL-1 receptor antagonist directly into the brain or peripherally [82,83]. More recently, Girard et al. used a rat model of perinatal brain injury combining prenatal LPS induced inflammation and postnatal hypoxia-ischemia [84]. The animals treated with the IL-1 receptor antagonist had both motor function and exploratory behavior preserved in the model. The myelin loss in the internal capsule and gliosis was also prevented in the animals that were treated [84]. Consistent with these findings, the neuroprotective effects of the immunosuppressive drug, minocycline, was thought to be mediated by the reductions in IL-1 synthesis after traumatic brain injury [85,86,87]. The findings in rodents summarized above emphasize the therapeutic potential of targeting the neuro-immune response to attenuate brain injury. The generation of mAbs is an effective and specific technique to block signaling pathways activated by a pro-inflammatory cytokine. Previous work has also demonstrated that anti-cytokine therapeutic strategies attenuate the effects of traumatic brain injury [88], stroke [54], and subarachnoid hemorrhage [89] in adult rodents.

Ovine-specific IL-1β and IL-6 proteins along with mAbs specific for these cytokines were successfully produced and purified [41,42]. The capacity of the mAbs to neutralize the cytokine inflammatory cascade was demonstrated in vitro in ovine splenic mononuclear cell cultures [43]. The pro-inflammatory effects of IL-1β and IL-6 proteins were characterized in mononuclear cells by the upregulation of the NF-κβ and STAT-3 transcription factors, respectively. NF-κβ and STAT-3 are downstream signaling mediators characteristically upregulated after IL-1β and IL-6 proteins bind to their receptors and activate pro-inflammatory signaling pathways [90,91,92]. The upregulation was reduced by treatment with the specific purified mAbs [43]. This in vitro study confirmed that anti-IL-1β and -IL-6 mAbs have high sensitivity and specificity for their corresponding ovine cytokine proteins. Based on these observations, we considered that these mAbs represented potential therapeutic candidates to attenuate the CNS inflammatory responses after ischemia in vivo. Consequently, we tested the ability of these mAbs to attenuate the disruption of the BBB and brain injury after exposure to in utero brain ischemia in fetal sheep.

### 3.2. Brain Distribution of mAb Antibodies

The delivery of antibody therapeutics into the brain to treat CNS disorders has represented a major challenge for drug development. In general, it has long been assumed that antibodies do not cross the intact BBB as a result of their large size and the absence of specific transporters. These factors severely restrict the targeting of specific molecules of interest within the brain via peripheral antibody administration because the efficacy of neuroprotective agents is strictly dependent on their ability to cross the BBB. 

Intravenous infusions of neutralizing anti-IL-1β mAb (approximately 5 mg/kg; two infusions over 2 h beginning 15 min and 4 h after 30 min of bilateral carotid artery occlusion) were administered over a 6-h interval after brain ischemia insult in fetal sheep. The mAb infusions were planned to accomplish early-sustained increases in systemic mAb concentrations in order to expose the cerebral microvasculature to mAb for an extended interval after ischemia. The mAbs were administered as systemic intravenous infusions because this is a clinically relevant route of administration and directly exposes the fetus to the mAb over an extended interval of time. Other routes of administration have previously been used to deliver antibodies directly into the brain via intracerebral injections in experimental studies of adult rodents [89,93]. Intracerebral injections of drugs facilitate the direct entry of drugs into the brain by circumventing the BBB. However, these routes of administration have much less relevance in the preclinical translational fetal sheep model and for the potential future treatment of human neonates. 

The mAb infusions resulted in sustained elevations in systemic mAb during the studies [50,94,95]. The infusions also resulted in significant anti-IL-1β mAb uptake into the fetal brain parenchyma and in the cerebrospinal fluid 24 h after the insult [50]. Therefore, the I/R insult facilitated mAb penetration into the fetal brain. The mAb may enter the fetal brain by crossing the injured fetal BBB after brain ischemia. A low rate of therapeutic mAb passage across the BBB is consistent with several studies that showed small amounts of brain uptake of antibodies against Aβ protein that was used to treat Alzheimer disease [96,97]. The uptake could have been enhanced because of ischemia-related damage to the cerebral vasculature [51]. Endothelial damage can facilitate diffusion into the brain by extracellular pathways but can also reduce the efflux from the brain to blood by receptor-mediated transcytosis. Efflux from the brain tissue via reverse transcytosis across the BBB has been previously reported to occur for immunoglobulins [98,99].

### 3.3. Neuroprotective Effects

The infusions of the neutralizing anti-IL-1β mAb after ischemia reduced the I/R related increases of IL-1β proteins within the brain parenchyma [50]. In addition, the mAb infusions were accompanied by small increases in TNF-α protein expression. This finding emphasizes the potential interactions among pro-inflammatory cytokines and suggests that by acting on IL-1β signaling, the systemically infused anti-IL-1β mAb could also have affected multiple neuro-immune responses within the ischemic brain [50]. More recently, we have shown that systemically infused anti-IL-1β mAb also decreases the transport of the IL-1β cytokine protein across the BBB after ischemia, at least in part by complexing with free IL-1β in the systemic circulation in fetal sheep [95]. Altogether, anti-IL-1β mAb infusions decrease IL-1β bioavailability in the fetal brain after I/R by (1) preventing IL-1β up-regulation in the brain parenchyma and (2) reducing the uptake of IL-1β across the BBB.

The most important finding was that systemic administration of the anti-IL-1β mAb attenuated I/R-related increases in BBB permeability across the brain regions measured by an inert non-specific molecule [50]. In addition, the BBB permeability showed an inverse linear correlation with concentration of the anti-interleukin-1β mAb in the parietal cortex. In summary, systemic infusions of anti-IL-1β mAb after ischemia result in anti-IL-1β mAb uptake into the brain, reduce I/R-related increases in the IL-1β protein, and promote increases in non-specific BBB permeability across brain regions in the fetus. 

Therefore, this therapeutic strategy has the potential to attenuate ischemia-induced neuro-inflammation and BBB disruption. This treatment could represent an important anti-inflammatory neuroprotective strategy to attenuate parenchymal brain injury after ischemia because both damage to the BBB and neuro-inflammation represent fundamental mechanisms of brain injury after ischemia. In addition, we have recently demonstrated that the anti-IL-1β mAb attenuates short-term histopathological I/R-related tissue injury, reduces ischemia-related increases in apoptosis, and reduces I/R-related increases in caspase-3 activity in the fetal brain [94]. The ischemia-related increases in apoptosis were predominately diminished in non-neuronal cells after treatment with anti-IL-1β mAb, suggesting that the mAb exerted its effects mainly on glial cells. Consequently, systemic infusions of anti-IL-1β mAb also attenuate short-term I/R-related parenchymal brain injury after ischemia in the fetus. The precise molecular mechanisms of action are complex, involving multiple cell types, and require further investigation. 

Similar to the studies described above, we have also shown that systemic infusions of anti-IL-6 mAb resulted in increases in mAb in plasma, brain parenchyma, and cerebrospinal fluid, and decreased IL-6 protein expression in the brain of a fetal sheep after ischemia [49]. The anti-IL-6 mAb infusions also diminished the ischemia-related increases in BBB permeability 24 h after ischemic injury, modifying tight junction and plasmalemma vesicle protein expression in the fetal brain. Therefore, inhibiting the effects of the IL-6 protein with systemic infusions of neutralizing antibodies attenuated ischemia-related increases in BBB permeability by inhibiting IL-6 after ischemia [49]. The results of these studies suggest that the pro-inflammatory cytokine, IL-6, also contributes to impaired BBB function after ischemic injury in fetus, and that treatment with an anti-IL-6 antibody could protect the developing fetal brain and provide a preventive and/or therapeutic strategy for ischemic brain injury in the perinatal period. 

In the studies summarized above, the fetal sheep was exposed to 30 min of carotid occlusion followed by mAb infusions over 6 h after brain ischemia [49,50,94,95]. The goal of this schema of administration was to produce prompt and sustained increases in circulating mAb to facilitate continued mAb availability to the cerebral vasculature and brain parenchyma. However, the precise timing of events resulting in perinatal brain injury can rarely be determined. Hence, mAb administration shortly after an adverse perinatal event would rarely be feasible. Consequently, additional research is required to calculate an accurate half-life of anti-cytokines mAbs and to determine the potential effects of more delayed treatment on perinatal brain injury before anti-cytokine mAbs can be considered as prevention or treatment strategies for perinatal ischemic brain injury. 

Furthermore, we have mostly focused on short-term outcomes of the anti-cytokine treatment strategies. Therefore, it is critical to investigate the neuro-protective efficacy of anti-cytokine therapies on long term outcomes after brain injury because persistent inflammation has been proposed to represent a tertiary phase of HI injury, which may further exacerbate injury and result in adverse outcomes later in life [100]. In addition to the potential long-term beneficial effects of anti-cytokine therapy, the potential adverse effects on the systemic immune system need to be considered. The neuro-immune response is a complex balance between pro-inflammatory and anti-inflammatory mediators, damaging effects, and repair processes. Even though pro-inflammatory cytokines are involved in the pathophysiological pathways resulting in neonatal brain damage and treatment with anti-cytokine antibodies exhibits anti-inflammatory effects, the expression of IL-1β is physiologically expressed during normal brain development and in vitro evidence suggests that the upregulation of IL-1β after an ischemic insult could be part of a protective response [101,102,103]. In addition, IL-1 signaling mediates the ability of oligodendrocytes to produce pro-angiogenic MMP-9 in vitro and in vivo, which facilitates angiogenic recovery after focal brain injury [104]. Likewise, IL-6 could act as a pro-inflammatory mediator during the acute phase of brain injury but also has neurotrophic properties during recovery from brain injury [105]. Therefore, the timing and duration of any anti-cytokine treatment needs to be carefully considered with respect to the timing of the initial brain injury. Nonetheless, most in vivo experiments have reported neurotoxic rather than neuroprotective effects of IL-1β and IL-6 in the developing brain in most instances during the acute phase of HI injury [13,14,15,16].

The brain ischemia was studied in an ovine fetus at 85% of gestation in our studies [49,50,94,95]. It is important to emphasize that the sheep brain at this time in gestation is generally considered to be similar to the brain of a near term human infant [32,39]. Therefore, it would also be important to examine the neuroprotective effects of the anti-cytokines mAbs in fetal sheep earlier in gestation to examine the potential beneficial effects and safety of this therapy in the premature brain after HI brain injury. 

Sexual dimorphism in response to neonatal HI is an important question that needs to be considered. In the human infant, male neonates exhibit more severe brain lesions after HI, resulting in more severe cognitive and motor outcomes than in their female counterparts [106]. Sex differences have been confirmed in rodent models of HI and in in vitro models of hypoxic cell death [107,108,109,110,111]. Although the molecular mechanisms of this sexual dimorphism remain largely unexplained, growing evidence suggests that inflammatory pathways may be one of the key players [111,112]. Therefore, this sexual dimorphism could be assessed in the fetal sheep model of I/R, as well as a potential sex-depend response to treatment with anti-cytokines mAbs.

Hypothermia is the only approved FDA therapeutic strategy to attenuate HIE in full term infants. However, this treatment is only partially effective, resulting in a rate of death or disability after treatment with hypothermia that from ranges from 31% to 55% in the reported trials [6,7,113]. Therefore, adjuvant therapeutic strategies are urgently needed [6,7,8,9]. In this regard, the addition of an anti-cytokine therapeutic treatment strategy could potentially augment the neuroprotection provided by hypothermia alone and further improve outcomes in infants exposed to HIE. However, future preclinical studies would be needed to evaluate the neuroprotective potential of hypothermia combined with a potential anti-cytokine therapy such as intravenous treatment with anti-cytokine mAb. 

## 4. Conclusions

Hypoxic-ischemic events in the preterm brain can initiate extensive inflammatory responses over a few hours that continue for days to weeks after the initial insult. The inflammatory response is characterized by the systemic and local release of pro-inflammatory cytokines that trigger alterations in BBB function and accentuate parenchymal injury. The results of our recent work in the fetal sheep brain strongly support the concept that neuro-inflammation represents a major mechanism in the brain injury and BBB disruption after I/R and, consequently, an important potential therapeutic target. In an effort to develop new therapeutic strategies to attenuate brain damage in neonates, we generated several anti-cytokines mAbs. After systemic infusions of the mAbs into fetal sheep with I/R brain damage, we reported that attenuated brain injury was associated with modulation of the neuro-immune response and improvement in BBB function. The results of our studies after I/R brain injury and systemic treatment with anti-cytokines mAbs are summarized schematically in Figure 2. We conclude that treatment with anti-cytokines mAbs protects the developing fetal brain and may provide an effective prevention/treatment strategy for perinatal ischemic brain injury.

## Figures and Tables

**Figure 1 brainsci-08-00101-f001:**
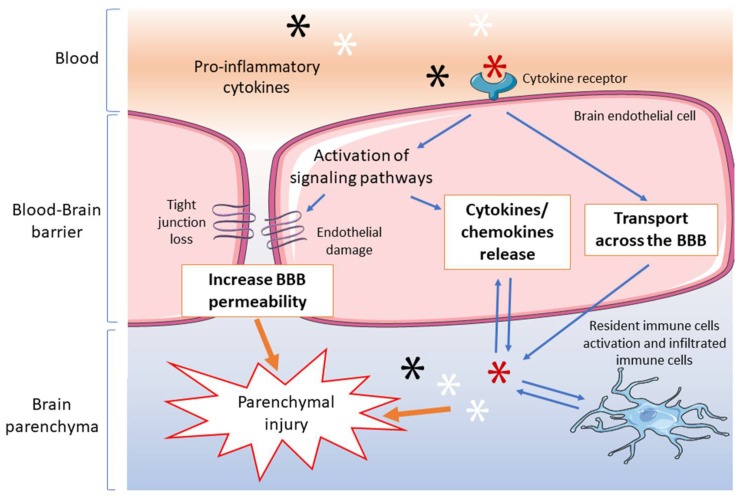
Schematic diagram of the interactions between the blood-brain barrier and the pro-inflammatory cytokines. Stars schematically represent cytokines. Circulating pro-inflammatory cytokines can activate signaling pathways leading to blood-brain barrier dysfunction, increase inflammatory responses, and be transported across the barrier by interacting with their receptors at the endothelial cells surface. Altogether, these signals at the blood-brain barrier can trigger parenchymal brain injury. Resident immune cell activation is most likely an important component of this response in the brain.

**Figure 2 brainsci-08-00101-f002:**
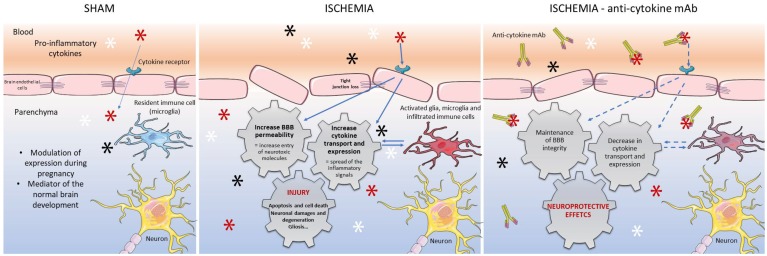
Schematic representation of the blood-brain barrier in the sham, ischemic-reperfusion, and in the ischemic—anti-cytokine mAb treated fetal sheep. Stars schematically represent cytokines in the sham and ischemic conditions; Ischemia—anti-cytokine mAb show cytokines (stars) complexed with the mAb in the blood and brain parenchyma. Systemic infusions of anti-cytokines neutralizing antibodies reduce ischemia-related blood-brain barrier dysfunction, transport of blood born cytokines, and cytokine expression within the brain to exert neuroprotective effects.

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
