# Peer review of "Anti-Cytokine Therapy to Attenuate Ischemic-Reperfusion Associated Brain Injury in the Perinatal Period"

_brainsci, 2018, doi:10.3390/brainsci8060101_

Round 1

Reviewer 1 Report

This review summarises the groups studies of using anti-cytokine monoclonal antibodies for hypoxic-ischemic injury in a fetal sheep model.

Well written but this review would benefit from subheadings in sections as it is easy to get lost in the text without them: Section 2 - cellular expression, BBB; etc Section 3; drug delivery; how treatment works, etc

Section 2 - 

More details about IL-1B and IL-6 generally, in development and in the brain would be useful - particularly microglia which is barely mentioned

It would be useful to mention inflammatory responses in the UCO (asphyxia) in near term fetal sheep model. Whilst not specifically H-I; umbilical cord occlusion still results in hypoxia, systemic ischemia and therefore is still relevant.

Section 3 - Impact on cell types more clearly defined.

There is a mention of adult rodent HI using these antibodies, to be more comprehensive discuss the use of anti-cytokine antibodies as a therapy, why these rather than anti-inflammatory medications?

Potential benefit of using with hypothermia?

The figures are useful but need a clear definition/legend of all the cell types and symbols

Author Response

this review would benefit from subheadings in sections as it is easy to get lost in the text without them: Section 2 - cellular expression, BBB; etc Section 3; drug delivery; how treatment works, etc

Response:  Subheadings have been added in sections 2 and 3 as follows:

2. Inflammation associated with ischemia-reperfusion injury in the ovine fetal brain

2.1. Cytokines expression in the developing brain

2.2 Blood-brain barrier dysfunction

3. Anti-cytokines antibodies as a potential neuroprotective strategy

3.1. anti-cytokines strategy

3.2. Brain distribution of mAb antibodies

3.3. Neuroprotective effects 

Section 2 - 

1)    More details about IL-1B and IL-6 generally, in development and in the brain would be useful - particularly microglia which is barely mentioned

Response: (Lines 114-124) We have emphasized the importance of astrocytes and microglia in the cytokine response.

2)    It would be useful to mention inflammatory responses in the UCO (asphyxia) in near term fetal sheep model. Whilst not specifically H-I; umbilical cord occlusion still results in hypoxia, systemic ischemia and therefore is still relevant.

Response: (Lines 125-130) Studies regarding umbilical cord occlusion in the fetal sheep model have been added:

“In the ovine model of repetitive umbilical cord occlusions, similar inflammatory responses were observed in the fetal brain.  IL-1β expression was increased in blood up to 24 h after exposure to repetitive umbilical cord occlusions and microglia counts were increased in the white matter and hippocampus at 24 h [55].  Other studies also demonstrated central activation of microglia associated with systemic inflammation and influx of neutrophils into the brain after 25 minutes of umbilical cord occlusion in fetal sheep [56-59].”

Section 3 - Impact on cell types more clearly defined.

3)    There is a mention of adult rodent HI using these antibodies, to be more comprehensive discuss the use of anti-cytokine antibodies as a therapy, why these rather than anti-inflammatory medications?

Response: (Lines 213-216)  Targeting inflammation using mAb have the advantage to be very specific to one signaling pathways. Given the dual role of many cytokines it is interesting to act on only one detrimental pathway.

The generation of mAbs is an effective and specific technique to block signaling pathways activated by a pro-inflammatory cytokine. Previous work has also demonstrated that anti-cytokine therapeutic strategies attenuate the effects of traumatic brain injury [88], stroke [54] and subarachnoid hemorrhage [89] in adult rodents.

4)    Potential benefit of using with hypothermia?

Response: (Lines 349-357)  We agree that there is a need to evaluate neuroprotection in addition to hypothermia treatment, this point have been added as a perceptive: “Hypothermia is the only approved FDA therapeutic strategy to attenuate HIE in full term infants. However, this treatment is only partially effective resulting in a rate of death or disability after treatment with hypothermia that from ranges from 31 % to 55 % in the reported trials [6, 7, 113]. Therefore, adjuvant therapeutic strategies are urgently needed [6-9].  In this regard, the addition of an anti-cytokine therapeutic treatment strategy could potentially augment the neuroprotection provided by hypothermia alone and further improve outcomes in infants exposed to HIE. However, future preclinical studies would be needed to evaluate the neuroprotective potential of hypothermia combined with a potential anti-cytokine therapy such as intravenous treatment with anti-cytokine mAb.”

5)    The figures are useful but need a clear definition/legend of all the cell types and symbols

Response:  We have revised figure 1 and 2 and missing legends for cells types and symbols have been added.

Reviewer 2 Report

This article summarized inflammatory response after hypoxia-ischemia brain injury and anti-cytokine therapy. It is a concise but informative review. Here are some suggestions:

Please specify the cellular resource of pro-inflammatory cytokines and the activities of brain immune-related cells(e.g., microglia, astrocyte) after HIE. 

White blood cells(e.g., neutrophils) play critical roles in the post-HIE inflammation and secretion of pro-inflammatory cytokines. Please discuss the activities of white blood cells after HIE. 

Except for system infusion, is there any other way to deliver anti-cytokine antibodies? Like the intracerebroventricular injection? If possible, please includes other delivery routes and compare them.

Author Response

1)    Please specify the cellular resource of pro-inflammatory cytokines and the activities of brain immune related cells (e.g. microglia, astrocyte) after HIE. White blood cells (e.g. neutrophils) play critical roles in the post-HIE inflammation and secretion of pro-inflammatory cytokines. Please discuss the activities of white bells after HIE.

Response: (Lines 113-121)  “We cannot be certain of the specific cellular origin of the cytokines in our studies.  However, microglia and astrocytes are the two major reactive glial cell types that play significant roles during CNS injury [13]. Pro-inflammatory cytokines are produced by both intrinsic and infiltrating immune cells. Circulating immune cells including neutrophils infiltrate the brain parenchyma to further exacerbate the inflammatory response and, consequently exacerbate brain injury. The infiltration of leukocytes is facilitated by chemokine secretion and expression of adhesion molecules at the endothelial cell surface.  Moreover, neutrophils that aggregate in the cerebral microvasculature, can interfere with cerebral circulation and potentially further exacerbate brain injury [16].”

2)    Except for system infusion, is there any other way to deliver anti-cytokine antibodies? Like the intracerebroventricular injection? If possible, please includes other delivery routes and compare them.

Response: (Lines 243-250)    The administration route is further discuss: “The mAbs were administered as systemic intravenous infusions because this is a clinically relevant route of administration and facilitates direct expose the fetus to the mAb over an extended interval of time.  Other routes of administration have previously been used to deliver antibodies directly into the brain via intracerebral injections in experimental studies of adult rodents [89, 93].  Intracerebral injections of drugs facilitate the direct entry of drugs into the brain by circumventing the BBB.  However, these routes of administration have much less relevance in the preclinical translational fetal sheep model and for the potential future treatment of human neonates.